# Deep learning-assisted co-registration of full-spectral autofluorescence lifetime microscopic images with H&E-stained histology images

Qiang Wang [1✉], Susan Fernandes[1], Gareth O. S. Williams[1], Neil Finlayson[2], Ahsan R. Akram [1], Kevin Dhaliwal[1], James R. Hopgood [3] & Marta Vallejo [4]

Autofluorescence lifetime images reveal unique characteristics of endogenous fluorescence in biological samples. Comprehensive understanding and clinical diagnosis rely on co-registration with the gold standard, histology images, which is extremely challenging due to the difference of both images. Here, we show an unsupervised image-to-image translation network that significantly improves the success of the co-registration using a conventional optimisation-based regression network, applicable to autofluorescence lifetime images at different emission wavelengths. A preliminary blind comparison by experienced researchers shows the superiority of our method on co-registration. The results also indicate that the approach is applicable to various image formats, like fluorescence in-tensity images. With the registration, stitching outcomes illustrate the distinct differences of the spectral lifetime across an unstained tissue, enabling macro-level rapid visual identification of lung cancer and cellular-level characterisation of cell variants and common types. The approach could be effortlessly extended to lifetime images beyond this range and other staining technologies.

[1] Centre for Inflammation Research, Queen's Medical Research Institute, University of Edinburgh, Edinburgh, UK. [2] Institute for Integrated Micro and Nano Systems, School of Engineering, University of Edinburgh, Edinburgh, UK. [3] Institute for Digital Communications, School of Engineering, University of Edinburgh, Edinburgh, UK. [4] School of Engineering and Physical Sciences, Heriot-Watt University, Edinburgh, UK. ✉email: Q.Wang@ed.ac.uk

Fluorescence lifetime imaging microscopy (FLIM) which can utilise lifetime contrast between healthy and pathological tissue has broad applications in biomedical diagnosis[1,2]. Without requiring the administration of exogenous biomarkers, autofluorescence lifetime imaging is of particular interest in clinical studies. Spectral histopathology for accurate diagnosis of lung cancer[3] and distinguishing of T-cell activation[4] are examples where FLIM images reveal the underlying metabolic state, pathological conditions, and the constitution of the samples associated with endogenous fluorescence. In general, FLIM images offer multi-dimensional information, including spatial, temporal, and spectral properties, where each dimension presents a unique perspective of the tissue under investigation. Conventionally, quantitative lifetime contrast of different tissues is usually identified by histogramming lifetime images to derive averaged lifetime, and qualitative comparison to the gold standard such as histology images is primarily based on visual inspection by human experts. However, accurate quantification and qualification depend on the reliable annotation of relevant histology images. In addition, the majority of existing technologies are effective at the macro level, where statistical information is the primary concern. This is often insufficient for cellular-level interpretation, such as cell types and sub-cellular components, which severely impedes the provision of transformative insight into fluorescence phenomena under investigation. To fill this gap, co-registration of FLIM and histology images can be used to gain an insightful understanding of the investigated tissue at both macro and micro levels and for revealing non-fluorescent features of the tissue and, therefore, limitations in the autofluorescence only approach.

However, co-registration remains difficult, especially for FLIM images at arbitrary emission wavelengths where particular structural features may not emit. This remains a challenge given the different nature of FLIM and histology images. First of all, there is a lack of statistical consistency in spatial sampling between image types for optimal transformations, as shown in Fig. 1 and Supplementary Fig. 1. Since fluorescence lifetime is independent of its intensity, lifetime images are visually much less structural but more homogeneous than histology images. A common practice to alleviate the homogeneousness is to adjust the saturation of each pixel in a lifetime image using the corresponding intensity for that pixel, as shown in the third row in Fig. 1. In addition, the emission spectrum of individual fluorophore is influenced by various environmental mechanisms and, hence, the images can be significantly different across wavelengths. The wavelength dependence of fluorescence decays represents an additional source of information about the underlying molecular environment. The dissimilarity of the image appearance and few explicit common features between lifetime

and histology images dramatically deteriorate the performance of conventional intensity- and feature-based co-registration. Secondly, the lack of availability of the ground truth for the co-registration impedes the application of many registration technologies. In this case, other more complex transformations, for example, homography would be required. Consequently, direct correlation of histology images with intensity/lifetime images, e.g., least-square[5], is not applicable, even when human intervention is introduced. Last, but not least, the preparation of tissue samples may introduce uncertainties. One common phenomenon is the colour variations, for example, due to the differences in staining and manufacturing of the scanners[6]. Meanwhile, artefacts may also be introduced during the preparation where structural changes[5] or contamination[7] of tissue may occur. Although various machine learning and deep learning (ML/DL) based approaches have been proposed to tackle the challenges in multi-modality image co-registration[8,9], straightforward applications of those methods may be infeasible, for example, because of the unavailability of ground truth and the visual contrast of the images. To the best of our knowledge, we did not find any prior work tackling this particular challenge to co-register FLIM images at arbitrary emission wavelengths with histology images.

Another potential solution is the direct translation from FLIM images to histology images to entirely bypass the co-registration. For example, Giacomelli et al.[10] proposed a virtual transillumination of epi-fluorescence multiphoton microscopic images to H&E-stained images of human breast tissue. Recently, contemporary DL technologies, particularly convolutional neural networks (CNNs), have achieved massive success in image-to-image translations where images in one domain are translated to another domain. Typical examples include, supervised methods, for instance, hyperspectral images to H&E-stained histology images[11], MedGAN[12] for multi-purpose medical image translation, translation from autofluorescence intensity images to histology images[5], semi/weakly/unsupervised methods, e.g., Cycle-MedGAN[13] for PET to CT image translation and MRI motion correction, and PC-StainGAN[14] to translate from H&E-stained to Ki-67-stained histology images. In general, the most frequently used architectures are UNet[15], generative adversarial networks (GANs)[16] and its variants, e.g., Cycle-GAN[17]. To ensure the quality of the translation, all methods require source and target images with competitive spatial resolutions, no matter whether they are tomographic or microscopic images. This, unfortunately, is often unavailable for FS-FLIM and histology images where often, for the reasons of image acquisition time and sample bleaching, the FLIM image is of greatly reduced spatial resolution. Therefore, direct applications may not be feasible. To allow the co-registration to be successfully achieved using FLIM images at arbitrary emission wavelengths within a specific range, we

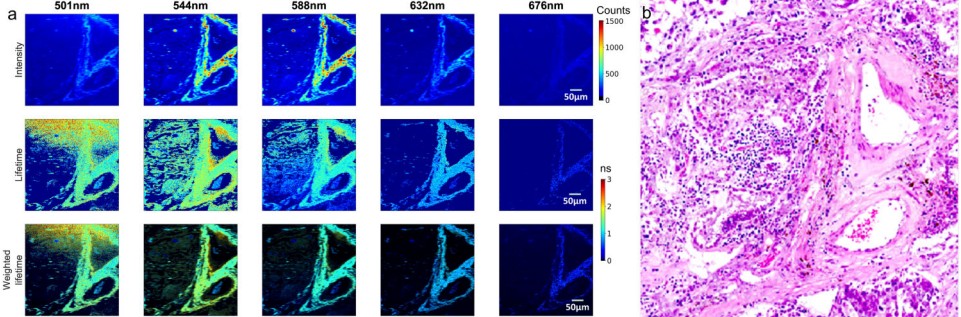

**Fig. 1 False-colour FS-FLIM images of 256 × 256 pixels, with a field of view (FOV) of 600 × 600 μm, at five different wavelengths. a** FLIM images at various emission wavelengths. The first row presents intensity images, the second row is the corresponding lifetime images, and the third row is the intensity-weighted lifetime images. **b** A manually cropped histology patch with a larger FOV than the FS-FLIM images.

propose a DL-assisted approach to overcome the aforementioned challenges. Full-spectral autofluorescence lifetime images at emission wavelengths across [500 nm, 780 nm] are collected via a custom-built ultra-sensitive FS-FLIM system[18]. Afterwards, they are translated into images similar in appearance to histology images, namely false histology images, using an unpaired image-to-image translation CNN model, where the translated images have a similar appearance to the original ones. Later, the corresponding histology patches are interactively cropped with human intervention from the whole slide images (WSIs), which are larger than those false histology images. Eventually, both images are input into a regression network, adapted from an intensity-based optimisation model, constrained by a partial photometric (PPM) condition. To evaluate the feasibility of the approach, FLIM images at different wavelengths and FOVs are input into the pipeline. In addition, various image modalities are tested, including intensity, lifetime, and their combination, to demonstrate the flexibility of the approach. The superiority of the translation is appraised by an ablation study, where results with/without the translation are blindly perceived by three experienced researchers with the fundamental knowledge of registration, FLIM and histology images. Finally, a direct application of the registration, i.e., stitching, is presented to illustrate the capability of stitched FLIM images at various wavelengths for rapid visual recognition of a human lung cancer tissue at a macro level.

## Results

**Registration**. The overall results are depicted in Fig. 2. All images were collected with a fixed spatial resolution of $256 \times 256$. The FOV of the images in the first row is $260 \times 260\,\mu m$, the second row is FOV $515 \times 515\,\mu m$, and the rest of the rows are $600 \times 600\,\mu m$. The first column represents the input images, where the first two images are set with a black background. The second column shows generated false histology images filtered by their corresponding intensity image as a mask. The third column corresponds to the real histology patches interactively cropped. The fourth column is the blending of the registration results per greyscale image, and the fifth column illustrates the blending of the false and real histology images of the registration results per the original colours of the false and original histology images. In addition to the lifetime images in the first three rows, intensity (fourth row) and intensity-weighted lifetime image (fifth row) are also evaluated. It is worth mentioning that the greyscale blending is visualised in green and magenta for FLIM images and histology images, respectively, to improve the visual presentation. This visualisation style is applied throughout the paper. It is also worth noting that due to the considerable discrepancies between the images, quantitative evaluation using conventional metrics may not correctly reflect the results. Therefore, whether or not the results are successful is primarily a subjective qualitative evaluation performed by human interpretation. Nonetheless, a quantitative comparison is still presented in Supplementary Table 1, where three similarity metrics, namely, mean squared error, normalised mutual information[19], and normalised cross-correlation[20] were calculated based on the registration outcomes on intensity, lifetime, and false histology images.

First of all, we appraise the impact of hardware configurations on the registration. In particular, we tried three lifetime images with different FOVs of $260\,\mu m$, $515\,\mu m$, and $600\,\mu m$, respectively and randomly selected wavelengths, as illustrated in Fig. 2. Because of the interactive cropping, the approach is able to generate effective registration regardless of the FOV. In addition, although the wavelengths of the first three rows are different, all are capable of producing reasonable results.

Secondly, RGB-colour images were also evaluated. In particular, three different visualisation results using the standard Jet colourmap were assessed. We first checked lifetime images with a dark background (the first and second columns in Fig. 2). Furthermore, we evaluated the images without the dark background on intensity (the fourth column in Fig. 2) and lifetime (the third column in Fig. 2). During the generation, we observed that the visualisation without the dark background might cause a blurred background in the generated false histology images. To ensure a successful co-registration, their corresponding intensity images need to be applied as a mask to the generated images. The last visualisation presentation is the aforementioned intensity-weighted lifetime images (the fifth column in Fig. 2). Similar to those images with the dark background, the masking is not required for the weighted lifetime images to achieve a qualitative co-registration.

We also evaluate the impact of image visualisation on the final results. Greyscale lifetime images were tested and the results are depicted in Fig. 3, which shows the feasibility of using greyscale lifetime as input. However, it is worth noting that to fulfil acceptable registration, the contrast of the original lifetime images needs to be enhanced. In this study, histogram equalisation technology[21] was applied to individual images. Meanwhile, the parameters of the regression need to be carefully tuned to achieve the target.

In addition, we also tried different modalities of images, including intensity and intensity-weighted lifetime. The fourth and fifth rows in Fig. 2 depict the corresponding intensity and intensity-weighted lifetime images. In order for the translation to be optimal, we trained the CycleGAN on the intensity and weighted images, respectively. The second column in Fig. 2 illustrates the translation results. Although the generated false histology images are visually different, all registrations using the false images present consistent results.

To further demonstrate the advantages of the proposed method, the regression was carried out in both greyscale and colour formats. The greyscale blending in the fourth column and the colour blending in the fifth column show that both formats can achieve the desired registration. Apparently, this is not feasible for lifetime images to utilise the original colours for the registration. The primary reason for the superiority of the colour image-based registration is the translation of lifetime images to false histology images. Visually, the overall appearance of the generated images is similar to real histology images, which implies that the values of the RGB channels are close enough for the regression to sort out an optimal homography estimation.

Finally, we compared the results of seven different wavelengths, including 500 nm, 526 nm, 552 nm, 578 nm, 605 nm, 631 nm, and 657 nm. In Fig. 4, all lifetime images are visualised with a dark background. The first row is the spectral lifetime images, the second row illustrates the generated false histology images, the third row shows the corresponding real histology patch with black background, and the fourth row depicts the greyscale registration results of blending the warped false histology images with the histology patch. At the shortest wavelength, e.g., 500 nm (first column in Fig. 4), the original lifetime image is relatively noisy due to a moderately low signal-to-noise ratio (SNR) and, hence, the corresponding generated image contains less structural information than others. Consequently, the registration needs to be carefully tuned, taking a relatively long time to obtain an optimal result. From the second to the fourth columns, the lifetime images are well reconstructed with visible structural content. As a result, the regression can be achieved robustly with a relatively short time for the optimisation, compared with other wavelengths. For the last three columns, the lifetime images become noisier and noisier with the increase of the wavelength, as

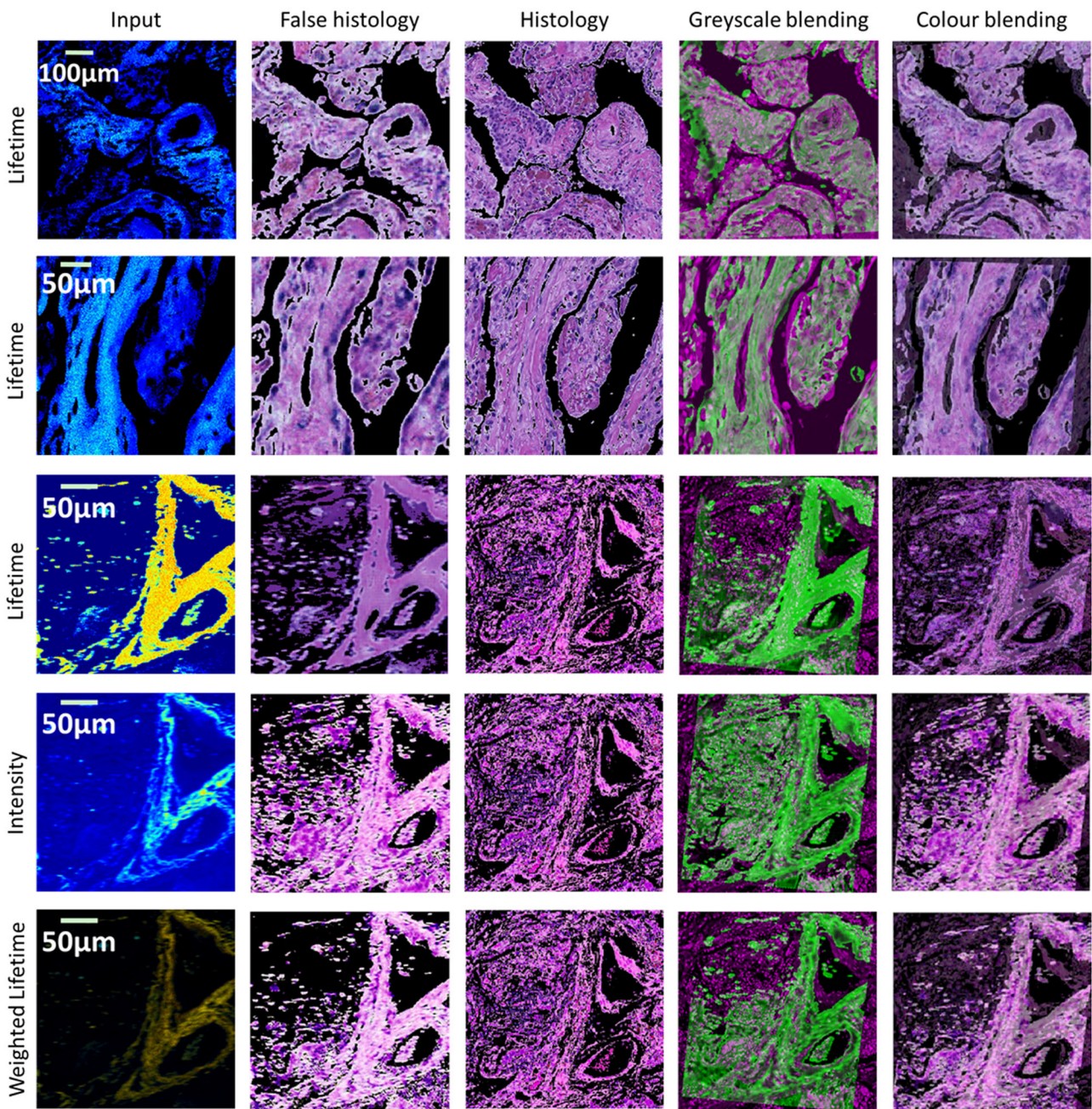

**Fig. 2 Registration results with the proposed approach.** All images are 256 × 256 in spatial resolution. The first row is a lifetime image at wavelength 616 nm with a FOV 260 × 260 μm. The second row is a lifetime image at wavelength 595 nm with a FOV 515 × 515 μm. The rest of the three rows are a lifetime, intensity, and intensity-weighted lifetime image, respectively, at 690 nm with a FOV 600 × 600 μm. The generated false histology images are presented (second column), along with the interactively cropped histology patch (third column). Registration results are illustrated by combining the false and real histology images together, using greyscale (fourth column) and the original colour(fifth column). To improve the visual interpretation, FLIM and histology images are visualised in green and magenta, respectively, in the greyscale blending (fourth column).

the decrease of the SNR. However, the predominant structural features suitable for qualitative co-registration are retained, and thus, an optimal registration can still be achieved.

**Comparison with/without translation**. To thoroughly evaluate the impact and necessity of the translated false histology images, we compared the registration with and without the CycleGAN, followed by a blind inspection performed by three independent researchers with some fundamental knowledge of FS-FLIM images and image registration. In particular, 40 lifetime images

with random wavelengths within the range of [500 nm, 710 nm] were selected from 40 hypercubes, which were sequentially acquired from a lung tissue sample. All lifetime images were visualised in greyscale with a fixed range, so that lifetime differences per wavelength were correctly reflected. All false histology images were converted into greyscale, without any contrast enhancement. During the regression, all parameters were fixed for both lifetime and false histology images, where the number of epochs was set to 200, the learning rate was initialised at 0.01 and decayed by 10 at epoch 100, and the window for PPM loss was 200.

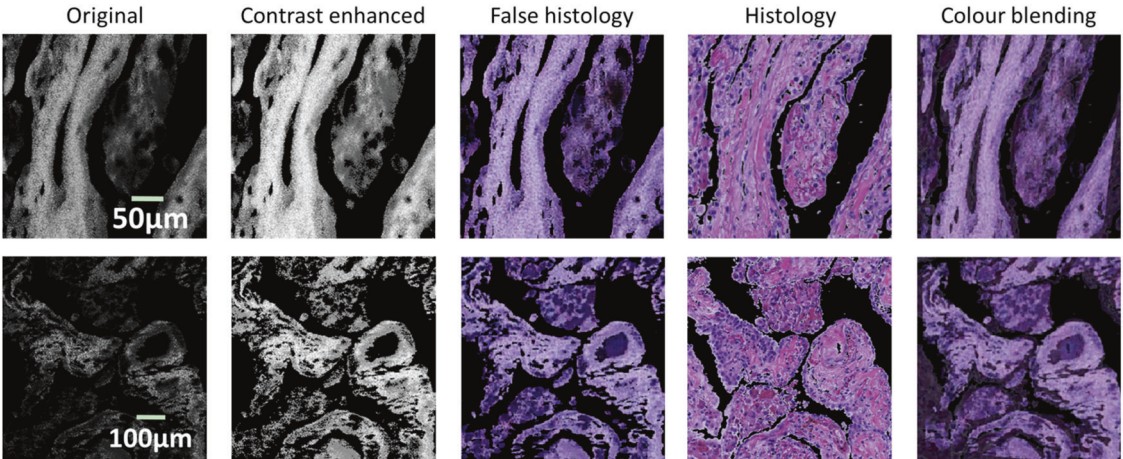

**Fig. 3 Example of the registration with lifetime images in greyscale.** The images in the first row correspond to the ones in Fig. 2. The original greyscale lifetime images (first column) need to be contrast-enhanced (second column) so that the generated false histology images (third column) and the corresponding histology patch (fourth column) can be reasonably registered (fifth column).

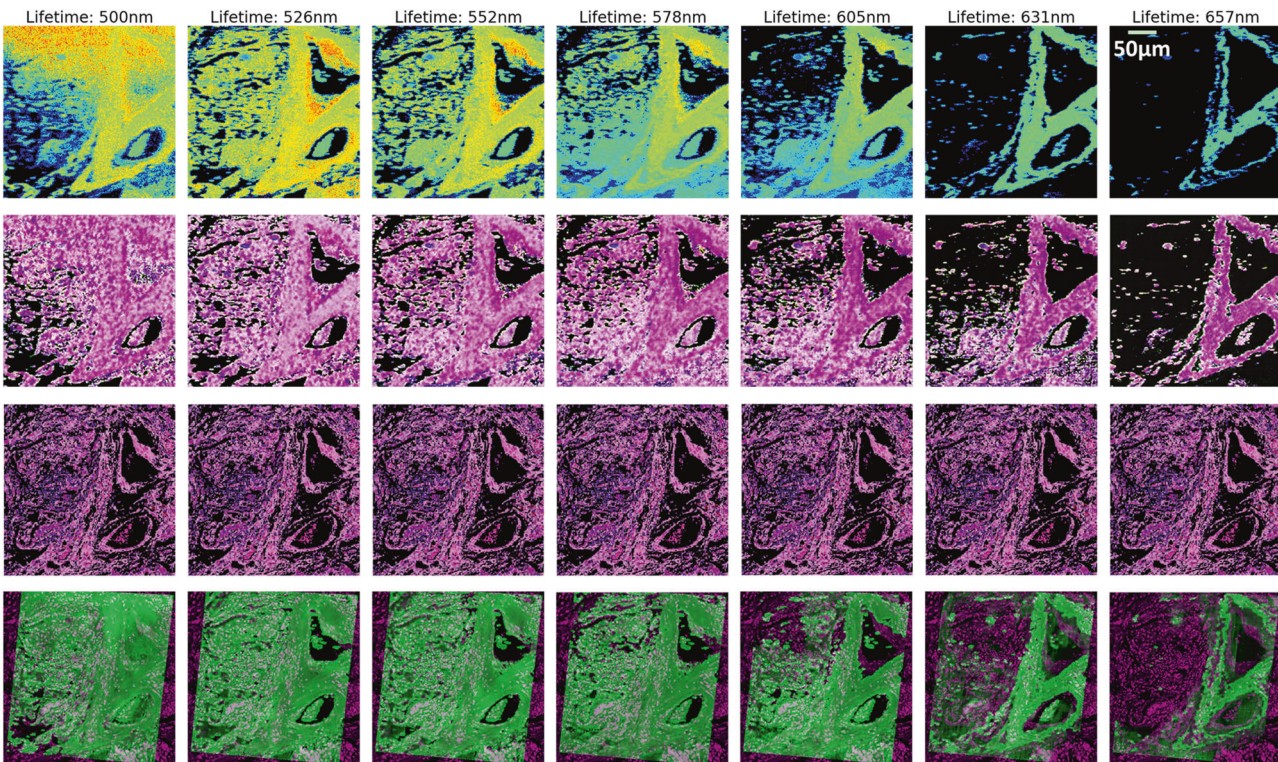

**Fig. 4 Registration results based on seven different emission wavelengths.** From rows 1–4, there are seven spectral lifetime images, the corresponding false histology images, the histology image patch, and the registered images in greyscale generated by blending translated false and real histology images, respectively.

Inspectors could choose one out of four options for each pair of registration results. That is, both images present good registration results, either false histology images or lifetime image performs well, and neither of them provides satisfactory outcomes. The inspection results are listed in Table 1, which suggests the superiority of the false images over the lifetime for the registration. On average, the proposed method achieves 67.5% satisfactory registrations, whereas the ablation without the translation reaches only 29.1% successful registrations.

To better understand the results listed in Table 1, we present four representative cases in Fig. 5, where all inspectors agreed on the choice for each case. The first row in Fig. 5 illustrates that both input images are able to contribute to plausible registration. The underlying reason for the success may owe to the structural resemblance between the input and the histology images. In the second case (second row in Fig. 5), the false image manages to recover some information that is hardly visible in the lifetime image, thus, resulting in a better registration performance. When taking a closer look at the lifetime and the result images, it seems that the upper half of the image could be registered with reasonable confidence. However, it is difficult to assess the lower part of the image as the presented information is insufficient. An interesting phenomenon occurs when the lifetime is better than the false histology image, as shown in the third row in Fig. 5.

Visually, the images present some structural information that is similar to the first case (the first row in Fig. 5), but the false image is unable to reach a satisfactory registration, whereas the lifetime is. We have checked the cases in this category, where all inspectors agreed, and we found that all those happened when background areas dominated the images, in particular at the edge of tissue samples. Given the FOV of the images and the FS-FLIM images presented, there are two (out of 40) such images presenting worse co-registration results and all these two images are at the right edge of the tissue samples. There are in total four

images at the right edge, where neither fake nor FS-FLIM images of the remaining two are unable to present satisfactory registration results. As a result, similar loss presented in Eq. (4) in the Discussion section might be derived from different homography transformations. As for the case when neither of the images successfully registers with the histology image (fourth row in Fig. 5), the lifetime image presents little meaningful information due to very low SNR, and the false image was unable to recover sufficient information. Consequently, the results were not meaningful, although the losses of the regression still converge.

In some extreme cases, particularly when the wavelengths were very long (over 740 nm), lifetime images (first column) struggled to present meaningful information, due to the high SNR. Supplementary Fig. 2 depicts four examples of these cases. In the first and second rows, the translated images (second column) appear to be structurally close to the lifetime images at shorter wavelengths, which is able to guarantee good registration results (fourth column). In other cases (third and fourth rows), although the translated images convey some structural information, it is insufficient to perform acceptable registration. The loss of information at a longer wavelength is expected, as the tissue sample is known to fluoresce mainly within the 500–650 nm range, with little to no emission expected beyond this.

**Table 1 Blind inspection of the registration results.**

|  | Inspector 1 | Inspector 1 | Inspector 1 | Average |
|---|---|---|---|---|
| Both satisfactory | 13 (32.5%) | 2 (5.0%) | 7 (17.5%) | 18.30% |
| false histology | 15 (37.5%) | 27 (67.5%) | 17 (42.5%) | 49.20% |
| Lifetime | 5 (12.5%) | 2 (5.0%) | 6 (15.0%) | 10.80% |
| Neither satisfactory | 7 (17.5%) | 9 (22.5%) | 10 (25.0%) | 21.70% |

Both results using the lifetime and generated false histology images at random wavelengths are presented to the inspectors to evaluate whether the registration is satisfactory or not.

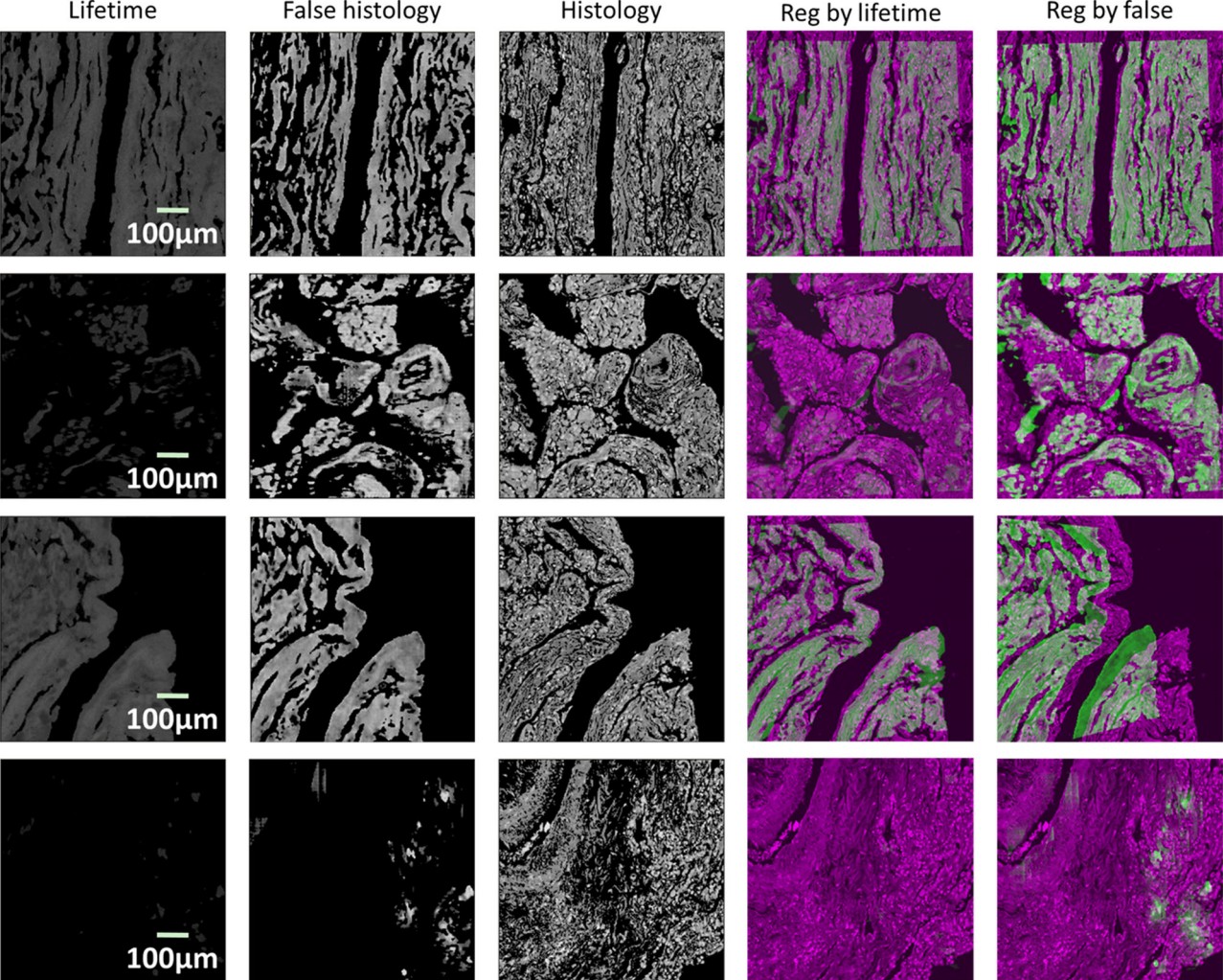

**Fig. 5 Comparison of the registration by the lifetime and false histology images.** Each row illustrates one specific case where both images present good registration results (first row), the false histology image outperforms the lifetime image (second row), the lifetime image is better than the false one (third row), and neither of them produces an acceptable result (fourth row), respectively.

Empirically, when the emission wavelength is less than 600 nm, FLIM images in greyscale may perform reasonably well for the co-registration without the translation, although sometimes a contrast enhancement algorithm may be needed. At wavelengths [600 nm, 650 nm], the enhancement is frequently required for a satisfactory co-registration. In addition, fine-tuning of the regression parameters is often needed, such as in Fig. 3. When the wavelength is beyond 650 nm, simply enhancing the contrast of the images may not be feasible for reaching a plausible co-registration, such as the one in the second row of Fig. 5. This may be due to the relatively low SNR, where "invisible" pixels contributing to the overall structure are filtered out during the contrast enhancement. It is worth mentioning that we have tried other multi-modality registration approaches on direct co-registration of the images, such as multi-scale intensity-based registration and elastic approaches[21]. Supplementary Fig. 3 demonstrates the results by a multi-scale intensity-based registration approach[22]. Despite exhaustive combinations of the parameters tried, a successful registration is unable to be achieved. This becomes better when using the generated false histology images for the registration, although the results are still not optimal, compared with the ones by the proposed method. Similar outcomes also happen to other advanced registration methods, such as elastic approaches. We believe that the primary reasons include the homogeneity of the FLIM images and the relatively low spatial resolution due to the tradeoff between the resolution and the data acquisition time for full-spectral FLIM images, where statistical consistency in spatial distribution of the images is challenging to be identified.

**Stitching**. A direct application of registration is stitching, where all individual images can be tiled up, forming a much larger image and, consequently, both local and global information can be revealed. In this study, we perform a simple strategy for the stitching of FS-FLIM images at various wavelengths. With the help of the developed software, the positions of the entire set of cropped histology patches were simply averaged. Figure 6 depicts an example of the stitching by reusing the data in[18], which contains 60 hypercubes collected across a human lung tissue sample. In order to maximise the visual contrast across the whole scanned area, the standard Jet colourmap was applied to allow more colours to be displayed. In addition, we enlarged the visible range of lifetime from [1.5 ns, 2.8 ns] to [1.0 ns, 3.0 ns] to further enrich the visual effect.

The histology slide has been interpreted by a lung pathologist, which demonstrates lung cancer (adenocarcinoma), tumour margin and adjacent healthy lung tissue. To improve the visual contrast and reflect sufficient structural information, intensity-weighted lifetime images are displayed, and the range of the lifetime is fixed for all visualised wavelengths at [1 ns, 3 ns]. The results are depicted in Fig. 6 and Supplementary Fig. 4, where six stitched lifetime images are presented at the wavelength of 500 nm, 555 nm, 582 nm, 609 nm, and 637 nm.

Compared with the histology images, the heterogeneous distribution of autofluorescence lifetime in the FS-FLIM images across the sample reveals distinct characteristics of the lung tissue under investigation. Lung cancer demonstrates a lower fluorescence intensity signal compared with healthy lung tissue, particularly at a shorter wavelength. Given a particular 527 nm wavelength (Fig. 6b), healthy pulmonary alveoli are clearly visualised and display a longer fluorescence lifetime (3 ns) compared with lung adenocarcinoma (1.5 ns). Interestingly, the walls of blood vessels and respiratory bronchioles display long fluorescence lifetimes, in keeping with healthy pulmonary alveoli, which may be related to the presence of elastin fibres. The

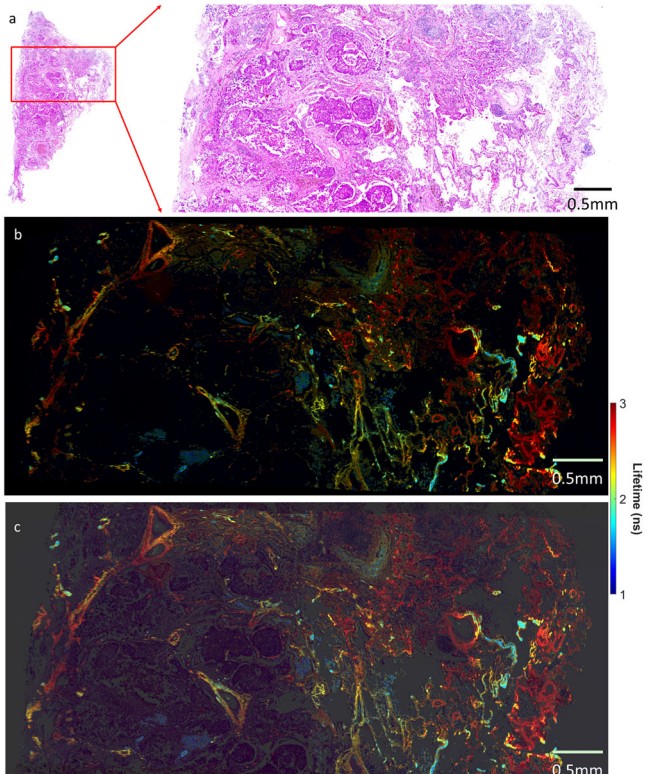

**Fig. 6 Stitching of sequentially acquired FS-FLIM images. a** is the corresponding H&E-stained histology image, where the enlarged area matches the ones by the FS-FLIM system. **b** is the stitching results per intensity-weighted lifetime images at wavelengths 527 nm. **c** is a blending of the weighted lifetime image and the corresponding histology image.

spectral lifetime across emission wavelengths also demonstrates the consistent decrease of the lifetime along with the increase of the emission wavelengths, which implies the potential for the lifetime-based differentiability of cancerous and non-cancerous tissue at wavelengths within the range [500 nm, 710 nm].

To reveal the fingerprint of individual cells from a lifetime viewpoint, six different types of cells were annotated at various locations in the histology image, including tumour, collagen, inflammation, stroma, red blood cell and alveolar septa, as shown in Fig. 7 (Source data is included in Supplementary Data 1). Due to the size difference between the cells and image pixels, lifetime values within $5 \times 5$ pixels were averaged as the absolute value of the cells. The lifetime of the cells is plotted at wavelength range [500 nm, 680 nm] in Fig. 7b. As mentioned previously, the histology image (Fig. 7a) demonstrates a transition from clinically confirmed lung cancer to healthy lung tissue from left to right. Tumour cells within the same area have similar lifetime, but they show a noticeable lifetime difference in different areas. For example, those in the transitional area (locations 4 and 5) have a longer lifetime than those in the cancerous area (locations 1, 2, and 3), which indicates they may be internally different. Alveolar septa also show a similar pattern, that is, normal cells (annotations 3 and 4) have a longer lifetime than those (annotations 1 and 2) in the transitional zone. This is also consistent with a visual inspection that the cells at positions 1 and 2 are thicker walled than the rest, suggesting a transitional change. In contrast, red blood cells (RBC) tend to have a consistent lifetime across the tissue. This is primarily because they are often in the vasculature and dominate the area. As for collagen, inflammation, and stroma, those types of cells in distinctive areas also have a different lifetime. However, due to

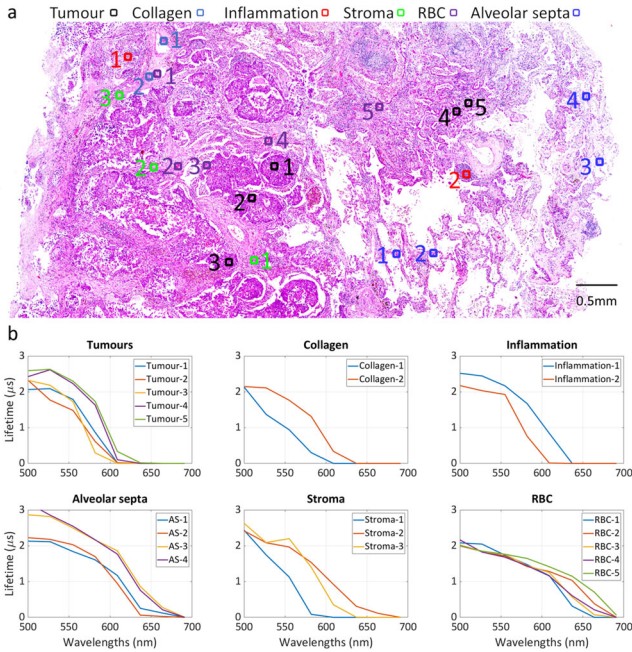

**Fig. 7 Absolute spectral lifetime of six different types of cells, including tumour, collagen, inflammation, stroma, red blood cell (RBC) and alveolar septa. a** Annotated locations of the cells are illustrated in the histology image. **b** The corresponding lifetime values for each cell type are presented across wavelengths in [500 nm, 680 nm].

the existence of other cells, their absolute lifetime is affected by the surrounding cells, and hence, shows less distinguishable features than other examined cell types.

In short, with the capability of the proposed registration, a pixel-level correlation between FS-FLIM and histology image can be built, and hence, comprehensive interrogation of FS-FLIM images at micro- and macro-level can be performed accurately and reproducibly. With the assistance of detailed annotation of the corresponding histology images, individual cells can be characterised using spectral lifetime. This enables quantitative differentiation of cell types and similar cells at different stages. With extra dimensions associated with FS-FLIM images, namely lifetime and spectrum, rapid and reliable recognition could be achieved over the histology images, requiring minimal expertise.

## Discussion

**FS-FLIM images**. Histology images are the "gold standard" for the interpretation of FLIM images, and therefore, pixel-level interpretation of FLIM images requires successful registration with histology images. Since FS-FLIM images consist of intensity and lifetime information at various wavelengths, both intensity and lifetime images at an arbitrary wavelength could be used for the registration. Ideally, intensity images seem more appropriate for that purpose, because both intensity and histology images are optically scanned to reflect concentration distribution. As the intensity images are a summation of the lifetime data, the use of the intensity image for registration implicitly provides for registration of the lifetime data to the histology image. Typical examples are those efforts on the virtual staining using auto-fluorescence intensity images for the registration with histology images[5,11]. However, the methods used may not be applicable to FS-FLIM with respect to wavelength images, where the quality of spectral intensity images is not comparable to histology images. Due to the flat nature of lifetime images, intensity-based images are anticipated to have better registration performance due to

increased structural information. However, as the intensity image and lifetime images are collected from the same acquisition, the successful registration of one image type provides registration of the other. We observed that intensity images might perform worse than lifetime images when used for co-registration, especially at long wavelengths, which may be due to the decrease in emission intensity in these regions. The evaluation metrics, shown in Supplementary Table 1, also demonstrate the same phenomenon. This is primarily because lifetime is independent of intensity. Low intensity may affect the quality of the images, but the corresponding lifetime in the images can still be reconstructed reliably. Consequently, the low-intensity areas are hardly visible in intensity images, whereas those areas are still present in the corresponding lifetime images, provided there is still a high enough signal to perform a lifetime calculation.

During the experiments, we observed that lifetime images are able to achieve reasonable registration results, using the proposed regression. For example, within the range of [520 nm, 600 nm], the quality of the images is sufficient to reflect some structural information, where similar features can be found in the histology images. However, post-processing on FLIM images for a direct registration may be required, such as contrast enhancement, so that the quality can be improved to match histology images. When the wavelength is outside of the range [520 nm, 600 nm], the image quality deteriorates, particularly when the wavelength is over 600 nm (Supplementary Fig. 1). In this case, lifetime images may not convey adequate features to fulfil the objective. Similar observations were also found on intensity image-based registration, but with a slightly narrower range of [530 nm, 600 nm].

**Generation of false histology images**. The Results section illustrated the success and necessity of the translation from lifetime to false histology images for the purpose of image registration. Due to the lack of availability of ground truth, the generation of false histology images needs to be based on unpaired image-to-image translation. This, however, helps the training of the model, where FS-FLIM and histology images do not need to be aligned. In practice, we trained the CycleGAN using histology images at various FOVs, including these relevant and irrelevant to the FLIM images. The generated results are still satisfactory for the registration purpose. Consequently, the translation enables the registration to be performed on images with similar appearance, allowing the regression on RGB colour images. In addition, the translation helps to recover "hidden" information in the original images, where the hardly visible part could be important but insufficient for the registration, such as in the second row of Fig. 5. With the assistance of the translation, both intensity and lifetime, even its combination can be utilised for the registration with plausible results. In some extreme cases, the translation is even able to recover structural content when the wavelength is over 740 nm, which is comparable to the high-quality FLIM images at short wavelengths (Supplementary Fig. 2).

In general, the inherent differences between histology images and FS-FLIM images results in imperfect translation. Consequently, the primary purpose of the translation is not to generate perfectly matched images. Instead, it enables the generated images to convey more features than FLIM images, which are more suitable for the registration. Apparently, better translation will result in more reliable and accurate registration. CycleGAN has achieved great success in style transfer between two separate domains, but the disadvantages are also significant when dealing with more complicated situations. For example, Liu et al.[14] proved that CycleGAN is not sufficient for the translation among different histology staining technologies due to the texture

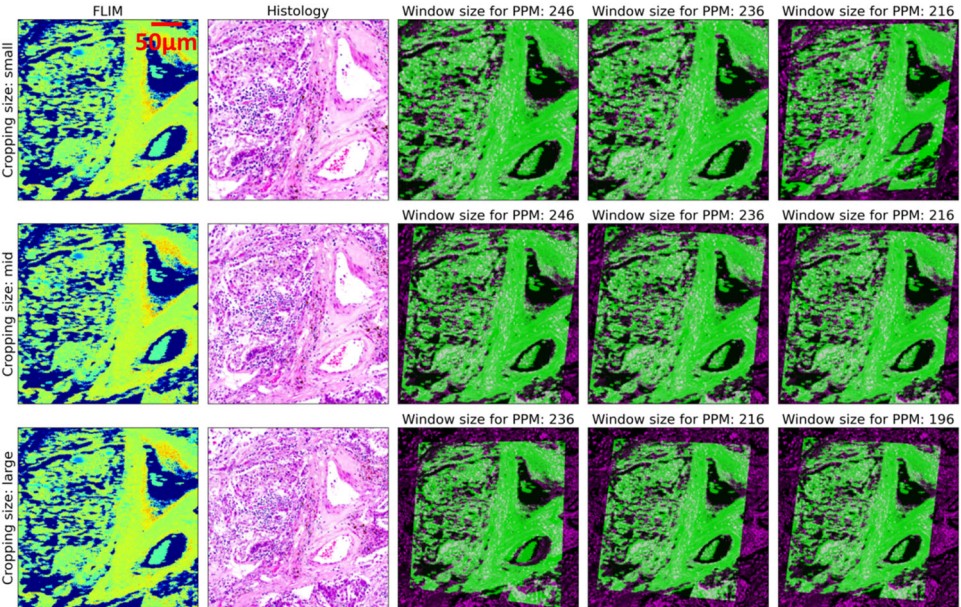

**Fig. 8 Impact of the cropping size of the histological patches and the window size for PPM on the results.** Three different sizes of the cropped histological patch are presented in each row, where "small" (first row), "mid" (second row), and "large" patch (third row) have a slightly smaller, similar, and larger FOV than the FLIM image, respectively. Three values of PPM are also evaluated and results are presented in columns three to five.

complexity. In this regard, extra improvements could be applied to our case to enhance the translation, such as texture loss[5] and the structural similarity constraint[14].

**Homography regression**. Due to the scale of microscopic images, slight differences among the orientation and position of the two modalities usually result in significant geometric changes. With the assumption that the underlying tissue samples are structurally consistent before and after the staining, homography transformation is, therefore, required to properly align the images. When input images or their features can be correlated flawlessly, various technologies could be applied for satisfactory co-registration, for example, directly linear transformation[23] and supervised[24] or unsupervised[25] homography estimation. Figure 2 and Supplementary Fig. 1 suggest that the FLIM and histology images used in this study are very different from each other in terms of FOV, structure, and colours, to name a few. Therefore, direct estimation of the homography matrix is one of the most effective and efficient ways for our case.

In addition, since the cropped histology patches are usually larger than lifetime images in regard to the FOV, the inclusion of large padding areas at the edge will also affect the regression. The ideal comparison would be only on the registered areas after transformation which, in practice, is not straightforward to achieve. Empirically, this can be approximated by a hollow rectangle mask, i.e., the PPM. During the experiments, we noticed that conventional metrics, namely, mean squared error and normalised mutual information, did not always reflect the registration performance correctly. That is, we observed that the proposed method presented a better result but worse metrics than with FLIM images only. Supplementary Table 1 shows that the registration with lifetime is better than the false histology image on normalised cross-correlation, but both our observation and the presented results demonstrate it is not the case. Again, the reason is probably because of the distinctions between the images. Alternatively, the PPM loss could be facilitated for an objective comparison. Unfortunately, we found that synthetic histology images always perform better than the lifetime images, in terms of

the PPM loss, even lifetime images achieved better co-registration than false histology images (Supplementary Fig. 6).

Since the regression model is a standalone module, it can be easily substituted by more advanced technologies for better estimation of the homography matrix. For example, a potential solution is the multi-scale image registration, where metrics at different scales are calculated and fused to obtain an optimal estimation[26]. It has also been reported that distortion between unstained and stained samples may be related to artefacts associated with sectioning and/or staining[7]. We also found a distortion effect in our study, which may be caused by the staining procedure and the confocal nature of the system employed as any structures at a depth not related to the focal plane will be lost. Nevertheless, non-rigid registration might be required, which could be supplemental to the regression or a method to replace it entirely.

Another potential artefact is the randomness of histology patches cropped manually by human. To evaluate the impact of the randomness on the results, we selected three different sizes of histology patches, which are smaller, similar, and larger than a given FLIM image, respectively. Since the window sizes for PPM loss also influence the regression, we repeated the co-registration of three histology patches with three window sizes. The results are depicted in Fig. 8. As far as histology patches are concerned, small ones (first row) require large window sizes, e.g., 246 and 236, to reach a satisfactory result. In contrast, large patches (third row) need a small window size for a reasonable registration, such as 216 and 196, where smaller window sizes do not satisfy it. For a suitable cropping with a slightly larger FOV (second row), different window sizes are able to generate sound outcomes. To achieve the optimal results, therefore, a slightly larger cropping with a moderate window size is recommended.

**Execution time**. An important aspect of our approach is the execution time. We evaluated this by repeating each step 10 times with different inputs, except the cropping of histology patches as it requires human intervention. Our testing platform is a desktop computer with eight-core CPU (3.0 GHz), 32GB memory, and

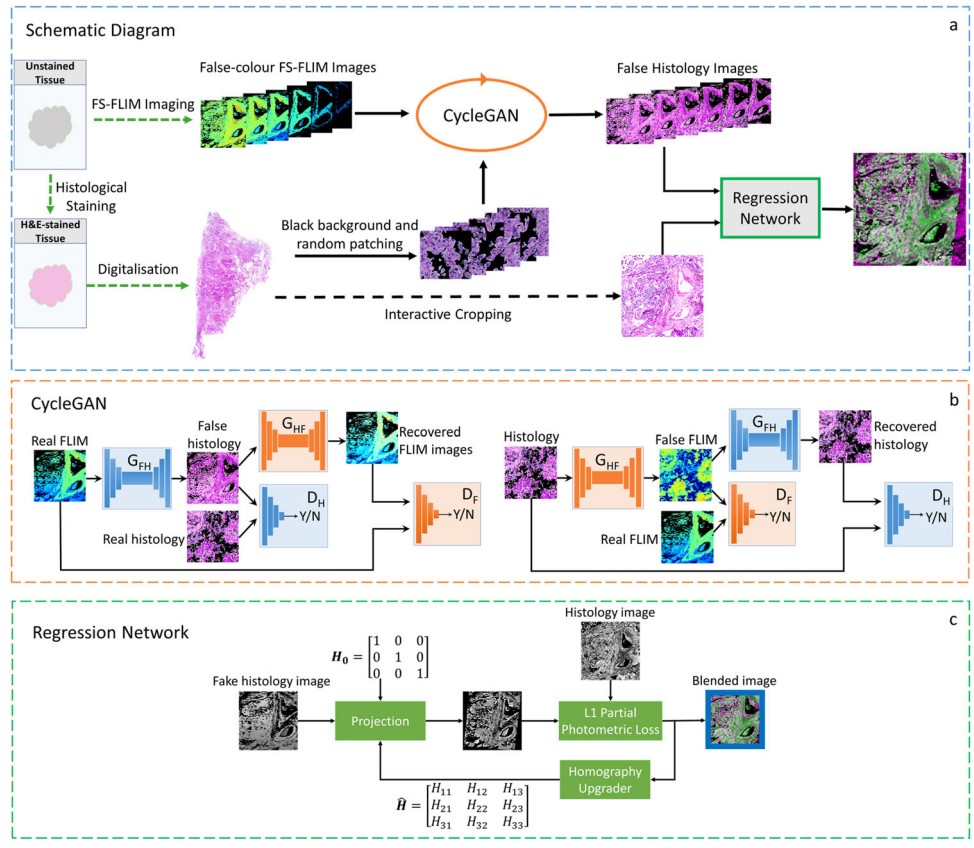

**Fig. 9 Schematic diagram of the proposed method. a** is the overall procedure of the method. After collecting the FS-FLIM images, they are input into the CycleGAN (**b**) to generate false histology images, in assistance with histology images. The output of the CycleGAN and the corresponding cropped histology patches are fed into a regression network (**c**) for the registration.

NVidia Titan RTX GPU (24GB memory). Note that the generation of false histology images and homography regression was performed on the GPU. Overall, the generation ran for 0.19 s, except for the first run which needed 20.52 s for initialising parameters and loading data to the memory. The regression configured with 200 epochs executed for 1.71 s in average, except for the first execution which needed 7.45 s.

## Methodology

The overall procedure of the proposed approach is depicted in Fig. 9, and it can be generally separated into three steps: data collection, false histology generation, and regression. FS-FLIM images were collected by a custom FS-FLIM system[18] on unstained tissue sections, and histology images were gathered by a bright-field microscope after the staining of the unstained tissue. After a simple data post-processing, both images were fed into the CycleGAN to generate false histology images. Later, the generated false histology images were input to a regression network, along with the corresponding histology patches cropped from the whole histology images to estimate the homography matrix. The detailed information about each step is discussed in the following sections.

**Data collection**. Ex-vivo human lung tissue samples were obtained from 11 patients with non-small cell lung cancer undergoing thoracic resection surgery, with paired non-cancer and lung cancer tissue sections obtained from each patient. Lung tissue specimens were fixed in 4% neutral buffered formaldehyde, and embedded in paraffin. Fixed unstained 5 μm slices were deparaffinised in xylene, rehydrated and subsequently dehydrated through gradient-ethanol, and mounted beneath coverslip.

Sequential 5 m sections were H&E-stained. All histology slides were digitally archived using AxioScan.Z1 slide scanner (Zeiss, Germany).

FS-FLIM images were acquired by a customised FS-FLIM system[18], which is capable of capturing time-resolved images over 512 spectral bands from 500 nm to 780 nm, and 32 time channels. The spatial resolution of the resultant images can be configured at different sizes. Due to the amount of data, we chose a size of $256 \times 256$ to find a trade-off between acquisition time and image quality. Consequently, a single measurement results in a 4-dimensional hypercube of $256 \times 256 \times 512 \times 32$, acquired with a 0.5 NA 20× objective (Olympus) and pixel dwelling time of 1 ms or 2 ms. Each hypercube contains multi-dimensional information in spatial, temporal, and spectral terms. The scanning was performed sequentially across the microarrays for all samples, resulting in a number of adjacent hypercubes, where the actual number depends on the size of each microarray, the configured FOV, and the overlapping between images. In total, we collected over 2,000 hypercubes on about 20 unstained tissue sections from several patients, using four different FOVs. It is worth mentioning that all of these FOVs lead to a significant undersampling of the samples.

**Generation of false histology images**. As far as the hypercubes are concerned, noise becomes noticeable at the edges of emission spectra due to the relatively low counts. To minimise the noise, a moving spectral mean of 8 channels (~4.5 nm) over this spectral region was deployed for lifetime estimation. Considering the amount of raw data, the computational resources available, and the reasonable quality of the images, a GPU-accelerated least-

square fitting[27] was utilised for the reconstruction of lifetime images. To reduce the photon quantum noise for optimal SNR, we used a threshold value of $\sqrt{\hat{N}}$ to further filter the images[28], where $\hat{N}$ is the mean of the measured fluorescence concentration. Similar to[29], the filter can be defined as:

$$\hat{i}^I_{x,y,s} = \begin{cases} 0 & i^I_{x,y,s} \leq \sqrt{\hat{N}} \\ i^I_{x,y,s} & otherwise \end{cases} \qquad (1)$$

$$\hat{i}^L_{x,y,s} = \begin{cases} 0 & i^I_{x,y,s} \leq \sqrt{\hat{N}} \\ i^L_{x,y,s} & otherwise \end{cases} \qquad (2)$$

where $I^I = \{i^I_{x,y,s} | i^I_{x,y,s} \geq 0, x, y \in [0, M] \, and \, s \in [0, S]\}$ is a full-spectral intensity image, and $I^L = \{i^L_{x,y,s} | i^L_{x,y,s} \geq 0, x, y \in [0, M] \, and \, s \in [0, S]\}$ is the corresponding lifetime image, with spatial size $M \times M$ and spectral size of S. Afterwards, a global normalisation is performed on the filtered data acquired on the same microarray to reflect the changes and consistency across each microarray and wavelength.

The FS-FLIM images use a dark background to indicate zero-value areas, whereas the histology images use a bright version. Accordingly, we need to mask the background of the histology images so that the values would not be misinterpreted during the generation. A simple approach, depicted in Supplement Fig. 5, is applied. The histology image is first converted to a greyscale image, which is further processed by inverting colour. Afterwards, histogram equalisation[21] is employed to enhance the contrast of the image, followed by the OTSU threshold selection method[30] to binarise the image. Eventually, the histology image with the black background can be derived by pixel-wise multiplication of the binary mask and the original histology image.

Due to the lack of availability of the ground truth for the translation from FLIM images to histology images, supervised methods are not applicable, and thus, unsupervised image-to-image translation technologies are considered. In this study, CycleGAN[17] is utilised for the generation of false histology images. Figure 9b shows the illustrative architecture of the CycleGAN, which contains two GANs: the transformation from FLIM images to false histology images $G_{FH} : I^F \rightarrow I^H$ (left half of the CycleGAN in Fig. 9) and the reverse transformation from histology images to false FLIM images $G_{FH} : I^H \rightarrow I^F$ (right half of the CycleGAN in Fig. 9), where $I^H$ and $I^F$ are FLIM and histology images, respectively. Furthermore, $D_{FH}$ and $D_{HF}$ are the discriminators associated with $G_{FH}$ and $G_{HF}$, respectively. The overall objective of the CycleGAN in this study can be defined as:

$$\begin{aligned} L(G_{FH}, G_{HF}, D_{FH}, D_{HF}) &= L_{FH}(G_{FH}, D_{FH}, I^F, I^H) \\ &+ L_{HF}(G_{HF}, D_{HF}, I^H, I^F) + \lambda L_{cyc}(G_{FH}, G_{HF}) \end{aligned} \qquad (3)$$

where $L_{FH}(G_{FH}, D_{FH}, I^F, I^H)$ is the adversarial loss of the mapping $G_{FH}$, $L_{HF}(G_{HF}, D_{HF}, I^H, I^F)$ is the adversarial loss of the mapping $G_{HF}$, $L_{cyc}(G_{FH}, G_{HF})$ is the cycle consistency loss, and $\lambda$ controls the weight of $L_{cyc}$.

For the training of the network, all FS-FLIM images are shuffled, regardless of the wavelength, and input into the network, which enables the translation at arbitrary wavelengths. Since it does not require paired images and the primary objective of this step is not to precisely transform FS-FLIM images into histology images, the transformation can be performed using the histology images irrelevant to the FLIM images. That is, arbitrary histology images of different sizes can be used for the generation of false histology images. In this study, about 40 WSIs were utilised for the generation. More specifically, 20 of the WSIs were scanned from the samples correlated with the FS-FLIM images, and the

rest were from the samples not related to the FS-FLIM images. The WSIs were cropped at random positions with different sizes from $256 \times 256$ up to $2048 \times 2048$, without considering the actual FOV of the patches. Those patches are later resized to $256 \times 256$ to be used as the input to the network.

During the training, the original CycleGAN was used and all hyperparameters were retained, except the batch size set to 16 and the epochs set to 50, since we observed that those values produced satisfactory results. The images in the second row in Fig. 4 illustrate the generated results for seven different wavelengths. Compared with the lifetime images (first row), the generated false histology images (second row) can recover some hidden information invisible in the lifetime images, particularly when the excitation wavelength is long. In addition, the appearance of the images shows similarities to real histology images.

**Regression network.** As mentioned in Section "Introduction", the primary challenges for the registration are the lack of ground truth and the nature of FS-FLIM and histology images. The widely-applied 8-point homography estimation[24,25] is not particularly applicable to our problem due to the unavailability of the targeting four points on histology images. In addition, features in FS-FLIM may be very different from those in histology images because of the differences among FLIM images at different wavelengths. Therefore, a direct way to tackle the challenges is to use conventional iterative regression. This study applies a simple yet effective iterative algorithm, where the homography matrix is directly estimated via a regression model. It is worth noting that while the homography matrix can be estimated by various methods, such as a 9-parameter direct vector, in this work the homography matrix is in the format of a $3 \times 3$ matrix to make the processing pipeline consistent as it is inherently facilitated in the libraries applied, such as OpenCV and Kornia.

Let $I^{F'}(x, y) = G_{FH}(I^F) = \{(x^{F'}_i, y^{F'}_i) | i \in (0, N-1)\}$ denote a false histology image, and $I^H(x, y) = \{(x^H_i, y^H_i) | i \in (0, N-1)\}$ the corresponding histology patch, where N is the dimension of the images after rescaling. The objective of the regression is to minimise the pixel-wise photometric using the L1 loss:

$$L(I^{F'}, I^H) = \frac{1}{N} \sum_{i=1}^{N} (|\mathcal{H}(I^{F'}(x_i, y_i)) - I^H(x_i, y_i)|) \qquad (4)$$

where $\mathcal{H}$ is the homography transformation, defined as:

$$\mathcal{H}(I^{F'}(x_i, y_i)) = \left\{ H \begin{bmatrix} x^{F'}_i & y^{F'}_i & 1 \end{bmatrix}^T | x^{F'}_i, y^{F'}_i \in I^{F'}(x_i, y_i) \right\} \qquad (5)$$

where H is the $3 \times 3$ homography matrix.

The procedure of the regression network is depicted in Fig. 9c. The regression starts with a $3 \times 3$ identity matrix $H_0$ to project the input fake histology image. Afterwards, photometric loss is calculated using Eq. (4) with L1 loss. Since the cropped histology patch has a different FOV from the false image, there will be some redundant information at the edge after the warping. During the experiments, we found that excluding the redundant edge area produces more robust results. Therefore, a PPM loss was applied, where the loss is derived, excluding the blue area of the blended intermediate result as it is seen in Fig. 9c, and optimised through gradient descent. To streamline the whole procedure and ensure optimal results, the regression network is adopted to allow all relevant operations to be directly performed on the Tensors of the image output from the CycleGAN.

**The software.** User-friendly open-source interactive software was developed to fulfil the aforementioned tasks, based on PyQt[31], OpenCV[21] for the image processing, Kornia[32] for the differentiable Tensor-based homography transformation, and PyTorch[33] for the CycleGAN and generation of the false histology images.

The source code is available on https://github.com/qiangwang57/coreg_flim_histology.

Supplementary Fig. 7 shows the software, which has four independent Graphic User Interfaces (GUIs). Supplementary Fig. 7a generates false histology images, where users can specify a FLIM image at a particular wavelength and the pre-trained parameters from the CycleGAN. To improve the quality of the generated images, the corresponding intensity can be utilised as a binary mask. Supplementary Fig. 7b allows users to load and crop a whole-side histology image to find the patch related to the particular FLIM image. The generated false histology image and the cropped histology patch are later fed into the regression GUI (Supplementary Fig. 7c), where a number of regression parameters, as well as different formats of the input images, can be adjusted to produce a reasonable regression result. Since a number of adjacent FS-FLIM images were collected per each microarray with a certain amount of overlapping at the edge areas, stitching is needed for both pixel- and global-level understanding of the given tissue samples in lifetime terms. Supplementary Fig. 7d enables users to perform the stitching based on the results from previous steps. The stitching results can be visualised in intensity, lifetime, or intensity-weighted lifetime at a specified wavelength, with/without the corresponding whole-slide histology image as the background. Supplementary Fig. 7d illustrates a stitching result using intensity-weighted lifetime without the background histology image.

**Statistics and reproducibility**. The CycleGAN was trained with unpaired FS-FLIM images with a fixed spatial resolution of $256 \times 256$ but various FOVs and histology images which were randomly cropped at various spatial resolutions from 40 WSIs. Since the patching of histology images for the regression needs human intervention, the cropped size impacts the regression results, where the patch with a similar FOV to the FLIM image is recommended for optimal registration outcomes. In addition, regression parameters should be also tuned to achieve satisfactory results.

**Ethics**. The samples used in this study were approved by regional Research Ethics Committee (NHS Lothian, Reference 15/ES/0094). All study participants gave written informed consent, and the study was conducted in accordance with the provisions of the Declaration of Helsinki.

**Reporting summary**. Further information on research design is available in the Nature Research Reporting Summary linked to this article.

## Data availability
All data associated with the results in this study are available on the University of Edinburgh DataShare facility (https://doi.org/10.7488/ds/3099 and https://doi.org/10.7488/ds/3421) and in Supplementary Data 1.

## Code availability
The lifetime reconstruction code is available on the University of Edinburgh DataShare facility (https://doi.org/10.7488/ds/3099). The source code of CycleGAN is available on https://junyanz.github.io/CycleGAN/. The software mentioned in the "The Software" section is available on https://github.com/qiangwang57/coreg_flim_histology.

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

## Acknowledgements

The authors acknowledge the financial support by the Wellcome Trust (grant number 206035/Z/17/Z) and the Engineering and Physical Sciences Research Council (grant number EP/S025987/1). S.F. is supported by the Medical Research Council (grant number MR/R017794/1). ARA is supported by a Cancer Research UK Clinician Scientist Fellowship (A24867). MV. is supported by the Engineering and Physical Sciences Research Council (grant number EP/K03197X/1 and EP/R005257/1). For the purpose of open access, the author has applied a Creative Commons Attribution (CC BY) licence to any Author Accepted Manuscript version arising from this submission.

## Author contributions

Q.W. proposed the ideas, conducted the experiments, developed the software, and prepared the manuscript. S.F. prepared, stained, and digitalised the lung samples, and drafted the first paragraph in Section "Data Collection". G.O.S.W. collected the data illustrated in the manuscript, and Q.W. collected the rest of data. N.F., Q.W., and G.O.S.W. developed the software and data analytics. Q.W. wrote the GPU acceleration code. S.F., A.R.A., and K.D. supplied the samples and provided valuable comments from the clinical perspective. J.R.H. and M.V. provided the comments from the perspective of signal processing and deep learning, respectively, to consolidate the ideas. All authors contributed to the manuscript. M.V., J.R.H., and K.D. supervised the project.

## Competing interests

The authors declare no competing interests.
