## [Peer Review File · Communications Biology]

Reviewers' comments:

Reviewer #1 (Remarks to the Author):

In this manuscript, the authors present a registration method for FLIM and histology images based on CycleGAN and homography matrix regression. The CycleGAN is used to generate false histology image from FLIM image, which is trained with unpaired image-to-image translation. The homography matrix for registration is then estimated via regression based on partial photometric (pixel-wise L1 loss). The method is interesting, however I have several questions regarding the methodology and experiment.

1. The network structure for the regression network is missing. The authors should include a figure and description in the main text about the regression network.
2. It also seems redundant to use a regression network to generate the homography matrix based on an 3x3 identity matrix. Directly using 9 parameters in the homography matrix for the optimizer would be a much straighter forward approach.
3. The authors claimed that "we noticed that conventional metrics, namely, mean squared error and normalised mutual information, did not always reflect the registration performance correctly" in line 353-355. How about the correlation between inspectors' judgement and the partial photometric loss used in the regression? The authors could present objective result using the partial photometric loss if it correlates better with the inspectors' judgement.
4. The experiment include an ablation study of the CycleGAN, but the method is not compared with any related works in literature. The authors could consider comparing to one of the other existing multimodal registration approaches to demonstrate the superiority of the proposed method.
5. The authors first claimed that "Since the regression model is a standalone module, it can be easily substituted by more advanced technologies for better estimation of the homography matrix. For example, the random sample consensus²⁵ is one of the most widely used and robust homography estimation methods." in line 360-362, and then claimed that "The widely-applied 8-point homography estimation^{23,24} is hardly applicable to our problem due to the unavailability of the targeting four points on histology images" in line 452-454. It's very confusing because RANSAC uses feature points based 8-point homography estimation method.
6. The greyscale blending in the figures are a little difficult to observe. The author could consider putting the two greyscale images in different RGB channels, so that only the overlapping part will be shown in grey. A reference image can be found in the link below.
(<https://www.researchgate.net/publication/308322242/figure/fig3/AS:413749744816130@1475657034412/Example-of-CT-T1-MRI-registration-of-the-brain-using-the-proposed-Evolution-method-Left.png>)

Reviewer #2 (Remarks to the Author):

This paper describes an approach for the co-registration of FS-FLIM images with the corresponding histology images. The paper is written well in a generally clear way. Their work consists of the following main components: 1) the collecting of FS-FLIM image; 2) fake histology image sequence generation from the FS-FLIM image sequence; 3) registration of fake histology image sequence to the corresponding manually identified real histology image patch; 4) subjective evaluation of the co-registration results; 5) stitching of the co-registered FS-FLIM images; and 6) software development with GUI. For components 2 (image translation) and 3(registration), they applied two well-known existing deep learning methods (CycleGAN and the regression network for homography estimation),

respectively. The novelty is not significant from the aspect of new machine/deep learning algorithm development. Given it is submitted to a biology journal and it is also very important to promote and demonstrate the strength of deep learning to the large biomedical community, I would like to focus more on the novelty from the application engineering aspect. That is, given the specific characteristics and challenges of the application at hand, selecting/applying/combining existing techniques to generate a new approach that are more suitable or better than existing approaches for addressing the specific task. Here are my questions/suggestions:

- 1) Is there any existing work in the literature on generating fake histology images from the FS-FLIM images using CycleGAN? Is there any existing work in the literature on using the regression network described in this paper to register FS-FLIM images and histology images?
- 2) What is the performance of existing co-registration approaches of FS-FLIM and histology images?
- 3) For the same FS-FLIM image sequence, what is the impact of different manual interactive cropping for the histology patch to the registration performance? (The cropping done by different people may be quite different for the same FS-FLIM image)
- 4) Which specific contrast enhancement method is used? Is it a global enhancement method or a local enhancement method?
- 5) In the experiments of "Comparison with/without translation" (Figure 5), what will it be if registering by using contrast enhanced lifetime instead of registering by using original lifetime? (It seems all the grayscale fake images are brighter than the corresponding original lifetime images)
- 6) For the case that lifetime gives better registration (such as row 3 in Figure 5), the authors say "We have checked the cases in this category, where all inspectors agreed, and we found that all those happened when background areas dominated the images, in particular at the edge of tissue samples." How many are such FS-FLIM images in the dataset? Is there any such FS-FLIM images that gets worse registration results than the fake images?
- 7) Will the authors make all the FS-FLIM images and histology images they collected publicly available? (The authors already made the software publicly available.)

Authors' response for: “Deep Learning-Assisted Co-registration of Full-Spectral Autofluorescence Lifetime Microscopic Images with H&E-Stained Histology Images”

Reviewers' comments:

First of all, we would like to appreciate the valuable comments from the reviewers to help us improve the quality of the manuscript. We have carefully considered all the comments one by one raised by the reviewers and we have revised the manuscript accordingly. Below we demonstrate how we have addressed the comments one by one in blue. We hope that our responses would sufficiently address the concerns raised by the reviewers.

Reviewer #1 (Remarks to the Author):

In this manuscript, the authors present a registration method for FLIM and histology images based on CycleGAN and homography matrix regression. The CycleGAN is used to generate false histology image from FLIM image, which is trained with unpaired image-to-image translation. The homography matrix for registration is then estimated via regression based on partial photometric (pixel-wise L1 loss). The method is interesting, however I have several questions regarding the methodology and experiment.

1. The network structure for the regression network is missing. The authors should include a figure and description in the main text about the regression network.

We appreciate the comment from the reviewer and we have updated Figure 9 on page 16 (which was Figure 8 in the original manuscript) and revised Section “Regression network” (line 483-493, page 18-19) with descriptive information regarding the network.

2. It also seems redundant to use a regression network to generate the homography matrix based on an 3x3 identity matrix. Directly using 9 parameters in the homography matrix for the optimizer would be a much straighter forward approach.

We agree with the reviewer that homography transformation can be achieved by various methods, and 9-parameter one is a straightforward method. The major reason that we use an 3x3 homography matrix is for the compatibility with third libraries used in the processing pipeline, such as Kornia and OpenCV, which also makes the implementation of our approach consistent. We have revised the first paragraph in Section “Regression network” (line 488-492, page 18) to emphasise our choice.

3. The authors claimed that “we noticed that conventional metrics, namely, mean squared error and normalised mutual information, did not always reflect the registration performance correctly” in line 353-355. How about the correlation between inspectors’ judgement and the partial photometric loss used in the regression? The authors could present objective result using the partial photometric loss if it correlates better with the inspectors’ judgement.

We appreciate reviewer’s advice on using partial photometric loss for objective comparison. We have investigated using this photometric loss function as suggested, but unfortunately, after further investigation, we found that this loss is also unable to reflect the registration results correctly. In fact, the loss from the proposed method is always higher than that based on lifetime images, even when lifetime image performs better than the corresponding false histology images. This may be due to the flatness of the lifetime images, which results in

significant differences between pixel values, whereas false histology images are much closer to the histology images in terms of pixel values. Two examples are presented in the following figures. The first one contains the case that both images can perform satisfactory registration results, whereas the second one shows when a lifetime image performs better than the false histology image, but still has a lower loss value. A new figure (Supplementary Figure 6) has been added to Supplementary information and the discussion has been appended to the first paragraph in Section “Homography regression” (line 378-381, page 14).

4. The experiment include an ablation study of the CycleGAN, but the method is not compared with any related works in literature. The authors could consider comparing to one of the other existing multimodal registration approaches to demonstrate the superiority of the proposed method.

We have tried different multimodal intensity-based registration approaches, such as multi-scale intensity-based registration (MMIR) [1] and elastic approaches, including bUnwarpJ [2], Fijiyama[3], and others in ImageJ. Unfortunately, we found that few can perform a satisfactory registration. The figure below depicts the results, where only registration by the MMIR is presented (as these are the best results of the other methods).

We believe that this is primarily because of the flatness of FLIM images, in comparison to the sharpness of histology images. In addition, the relatively low spatial resolution, due to the tradeoff between resolution and data acquisition time, also impacts the registration performance. Another possible reason is the different FOVs between FLIM and histology images, which makes the registration more challenging.

The description of the results has been added as the last paragraph in Section “Comparison with/without translation” (line 234-244, page 9) and the demonstration has been added as Supplementary Figure 3.

[1] <https://uk.mathworks.com/help/images/registering-multimodal-mri-images.html>. June 2022.

[2] <https://github.com/fiji/bUnwarpJ>

[3] <https://github.com/Rocsg/Fijiyama>

5. The authors first claimed that “Since the regression model is a standalone module, it can be easily substituted by more advanced technologies for better estimation of the homography matrix. For example, the random sample consensus²⁵ is one of the most widely used and robust homography estimation methods.” in line 360-362, and then claimed that “The widely-applied 8-point homography estimation^{23,24} is hardly applicable to our problem due to the unavailability of the targeting four points on histology images” in line 452-454. It’s very confusing because RANSAC uses feature points based 8-point homography estimation method.

We apologise for the mistake that included RANSAC as a potential solution for better estimating the homography matrix in our study. We have revised the relevant paragraph (line 383-384, page 14) to avoid confusion.

6. The greyscale blending in the figures are a little difficult to observe. The author could consider putting the two greyscale images in different RGB channels, so that only the overlapping part will be shown in grey. A reference image can be found in the link below.

(<https://www.researchgate.net/publication/308322242/figure/fig3/AS:413749744816130@1475657034412/Example-of-CT-T1-MRI-registration-of-the-brain-using-the-proposed-EVolution-method-Left.png>)

We appreciate the advice from the reviewer to improve the visual presentation and interpretation of the registration results and we have revised all figures accordingly, including Figures 2, 4, 5, 8, 9, and Supplementary Figures 2, 3, and 6.

Reviewer #2 (Remarks to the Author):

This paper describes an approach for the co-registration of FS-FLIM images with the corresponding histology images. The paper is written well in a generally clear way. Their work consists of the following main components: 1) the collecting of FS-FLIM image; 2) fake histology image sequence generation from the FS-FLIM image sequence; 3) registration of fake histology image sequence to the corresponding manually identified real histology image patch; 4) subjective evaluation of the co-registration results; 5) stitching of the co-registered FS-FLIM images; and 6) software development with GUI. For components 2 (image translation) and 3 (registration), they applied two well-known existing deep learning methods (CycleGAN and the regression network for homography estimation), respectively. The novelty is not significant from the aspect of new machine/deep learning algorithm development. Given it is submitted to a biology journal and it is also very important to promote and demonstrate the strength of deep learning to the large biomedical community, I would like to focus more on the novelty from the application engineering aspect. That is, given the specific characteristics and challenges of the application at hand, selecting/applying/combining existing techniques to generate a new approach that are more suitable or better than existing approaches for addressing the specific task. Here are my questions/suggestions:

1) Is there any existing work in the literature on generating fake histology images from the FS-FLIM images using CycleGAN? Is there any existing work in the literature on using the regression network described in this paper to register FS-FLIM images and histology images?

To the best of our knowledge, we did not find any reference working on the co-registration of FS-FLIM and histology images. As the reviewer mentioned, this study was not designed to develop novel DL techniques for the co-registration. Instead, it intends to find an effective way for the co-registration so that absolute lifetime signatures could be revealed for advanced characterisation of lung cancer, which is missing in the current context. We have revised the second paragraph in the introduction (line 69-71, page 3).

2) What is the performance of existing co-registration approaches of FS-FLIM and histology images?

As we mentioned, we have not found any methods concerning this particular problem. However, we have made a further comparison with existing co-registration approaches as proposed by Reviewer 1 (comment 4), such as multi-scale intensity-based registration and the results are presented in the last paragraph in Section “Comparison with/without translation” (line 234-244, page 9) and Supplementary Figure 3.

3) For the same FS-FLIM image sequence, what is the impact of different manual interactive cropping for the histology patch to the registration performance? (The cropping done by different people may be quite different for the same FS-FLIM image)

We agree with the review that the size of histology patches affects the registration. Given a window size for the partial photometric loss, the patches with a similar size to the given FLIM image usually have better performance than those with significantly different sizes. However, this can be overcome by larger window sizes for partial photometric loss. In general, the cropping close to the FLIM image with a moderate window size is able to achieve satisfactory results. A new paragraph has been added as the last paragraph in Section “Homography regression” (line 391-401, page 15), along with a new figure (Figure 8 on page 15) below to illustrate the comparison of different cropping and window sizes.

4) Which specific contrast enhancement method is used? Is it a global enhancement method or a local enhancement method?

Histogram equalisation provided by OpenCV [1] is used locally, rather than globally. The relevant paragraph (line 136-137, page 5) has been revised.

[1] OpenCV. Open Source Computer Vision Library. <http://opencv.org/>. Mar. 2021.

5) In the experiments of “Comparison with/without translation” (Figure 5), what will it be if registering by using contrast enhanced lifetime instead of registering by using original lifetime? (It seems all the grayscale fake images are brighter than the corresponding original lifetime images)

Yes, contrast-enhanced lifetime images can yield better registration performance. As discussed in the last paragraph in the section “Comparison with/without translation” (line 234-244, page 9) and the last paragraph in “FS-FLIM images” in “Discussion” (line 324-332, page 13), we observed that at wavelength range [520nm, 600nm], lifetime images contain sufficient structural information to perform reasonable registration, where contrast enhancement is not required. At a wavelength range [600nm, 650nm], contrast enhancement is always required for satisfactory co-registration. When the wavelength is longer than 650nm, contrast enhancement may not be feasible for plausible co-registration due to the relatively low signal-noise ratio.

6) For the case that lifetime gives better registration (such as row 3 in Figure 5), the authors say “We have checked the cases in this category, where all inspectors agreed, and we found that all those happened when background areas dominated the images, in particular at the edge of tissue samples.” How many are such FS-FLIM images in the dataset? Is there any such FS-FLIM images that gets worse registration results than the fake images?

We found that the number of such FS-FLIM images is quite small. Given the FOV of the images and the FS-FLIM images presented, there are two (out of 40) such images presenting worse co-registration results and all these two images are at the right edge of the tissue samples. There are in total four images at the right edge, where neither fake nor FS-FLIM images of the remaining two are unable to present satisfactory registration results. We have revised the paragraph (line 207-211, page 9).

7) Will the authors make all the FS-FLIM images and histology images they collected publicly available? (The authors already made the software publicly available.)

The images presented in the paper are open-sourced, which can be found on DataShare provided by the University of Edinburgh via <https://doi.org/10.7488/ds/3421> and <https://doi.org/10.7488/ds/3099>. The full dataset will be made publicly available once our current research on the dataset completes.

REVIEWERS' COMMENTS:

Reviewer #1 (Remarks to the Author):

Thanks the authors for revising the manuscript, and all my questions are addressed.

Reviewer #2 (Remarks to the Author):

Thank the authors for addressing all of my previous comments. I am satisfied with the responses. It would be good if they add information on how long it takes for running the whole process in their tool.

Authors' response for: “Deep Learning-Assisted Co-registration of Full-Spectral Autofluorescence Lifetime Microscopic Images with H&E-Stained Histology Images”

Reviewers' comments:

First of all, we would like to appreciate the valuable comments from the reviewers to help us improve the quality of the manuscript. We have carefully considered all the comments one by one raised by the reviewers and we have revised the manuscript accordingly. Below we demonstrate how we have addressed the comments one by one in blue. We hope that our responses would sufficiently address the concerns raised by the reviewers.

Reviewer #1 (Remarks to the Author):

Thanks the authors for revising the manuscript, and all my questions are addressed.

We appreciate the reviewer's comments that our revision is satisfactory.

Reviewer #2 (Remarks to the Author):

Thank the authors for addressing all of my previous comments. I am satisfied with the responses. It would be good if they add information on how long it takes for running the whole process in their tool.

To address the comment of the reviewer, we have added a new section “Execution time” as the last subsection of Section “Discussion”. Particularly, we have evaluated the execution time of each step involved in the procedure, except for the cropping of histology images as this is performed by human. Overall, the generation ran for 0.19 seconds, except for the first run which needed 20.52 seconds for initialising parameters and loading data to the memory. The regression configured with 200 epochs executed for 1.71 seconds in average, except for the first execution which needed 7.45 seconds.